# Evaluating the Accuracy of Privacy-Preserving Large Language Models in Calculating the Spinal Instability Neoplastic Score (SINS)

**DOI:** 10.3390/cancers17132073

**Published:** 2025-06-20

**Authors:** Li Yi Tammy Chan, Ding Zhou Matthew Chan, Yi Liang Tan, Qai Ven Yap, Wilson Ong, Aric Lee, Shuliang Ge, Wenxin Naomi Leow, Andrew Makmur, Yonghan Ting, Ee Chin Teo, Tan Jiong Hao, Naresh Kumar, James Thomas Patrick Decourcy Hallinan

**Affiliations:** 1Department of Diagnostic Imaging, National University Hospital, Singapore 119074, Singapore; tammy.chan@mohh.com.sg (L.Y.T.C.); matthew.chan@mohh.com.sg (D.Z.M.C.); yi_liang_tan@nuhs.edu.sg (Y.L.T.); wilson.ong@mohh.com.sg (W.O.); aric_lee@mohh.edu.sg (A.L.); shuliang_ge@nuhs.edu.sg (S.G.); andrew_makmur@nuhs.edu.sg (A.M.); yonghan_ting@nuhs.edu.sg (Y.T.); ee_chin_teo@nuhs.edu.sg (E.C.T.); 2Biostatistics Unit, Yong Loo Lin School of Medicine, Singapore 117597, Singapore; qaiven@nus.edu.sg; 3AIO Innovation Office, National University Health System, Singapore 119228, Singapore; naomi_wenxin_leow@nuhs.edu.sg; 4University Spine Centre, University Orthopaedics, Hand and Reconstructive Microsurgery, National University Health System, Singapore 119074, Singapore; jonathan_jh_tan@nuhs.edu.sg (T.J.H.); dosksn@nus.edu.sg (N.K.)

**Keywords:** large language models, radiology, artificial intelligence, spine metastases

## Abstract

Spinal tumours can result in spinal instability, and clinicians utilise the Spinal Instability Neoplastic Score (SINS) to determine if surgical intervention is required. However, the calculation of the SINS is time-consuming. Large language models may improve the calculation of the SINS, but their accuracy remains underexplored. The authors aim to evaluate two models—Claude 3.5 and Llama 3.1—against clinician assessments. The authors hope that the findings from this study may lead to the implementation of large language models in clinical practice to streamline workflows and improve consistency in the assessment of spinal metastases.

## 1. Introduction

In recent years, large language models (LLMs) have gained increasing accessibility, unlocking a wide range of applications within healthcare [1,2]. One of the key advantages of LLMs is their ability to handle repetitive, data-intensive tasks with precision and efficiency [3]. In the field of diagnostic radiology, LLMs have been tested in a large spectrum of clinical tasks from processing clinical request forms [4,5], automating the quantitative measurement of tumours [6,7,8] and the computation of clinical scoring systems [9,10]. By automating these routine tasks, LLMs can streamline clinical workflows, enabling clinicians to devote more time to complex decision-making and patient care [11,12]. However, as the adoption of AI in healthcare grows, so too does the need to address concerns regarding data security and the preservation of patient privacy. With the right safeguards in place, such as data anonymization and encryption, LLMs offer the potential to improve efficiency without compromising patient confidentiality [13,14].

The clinical utility of LLMs extends to oncology, particularly in the diagnosis and management of spinal metastases. One critical tool in assessing patients with metastatic spine disease is the Spine Instability Neoplastic Score (SINS) [15]. The SINS helps clinicians evaluate the risk of mechanical spinal instability and determines whether urgent surgical intervention is needed before initiating systemic treatments like radiation therapy or chemotherapy. Given that the SINS incorporates multiple variables, including clinical symptoms and imaging findings, the task of data extraction and score calculation can be time-consuming and subject to variability. This is where LLMs have the potential to improve accuracy and consistency in calculating the SINS [1,16].

While prior studies have evaluated the accuracy of generically trained LLMs in calculating clinical scores, their use in determining the SINS from clinical records remains unexplored [9,10]. This study aims to fill that gap by assessing the accuracy of SINS calculation using two institutionally available privacy-preserving large language models (PP-LLMs), comparing their performance directly against that of clinicians routinely involved in calculating these scores. A robust reference standard will be established in consensus with three experienced spine oncology clinicians. The analysis will integrate data derived from MRI and CT radiology reports, combined with patient pain histories documented in the electronic medical record (EMR). By evaluating the performance of institutional PP-LLMs, this study seeks to assess the potential for LLMs to assist clinicians in spine metastasis management, ensuring both improved diagnostic efficiency and the secure handling of sensitive patient data.

## 2. Methods

### 2.1. Study Design

This was a retrospective study conducted in a single centre (National University Hospital, Singapore). The study was granted a waiver from the domain-specific review board due to its low-risk nature as only de-identified data were used. All patients with spinal metastases who underwent both computed tomography (CT) and magnetic resonance imaging (MRI) of the spine between January 2020 and December 2022 were included. Patients with prior spinal instrumentation or with missing data (i.e., incomplete radiological reports or electronic medical record entries) were excluded. All CT and MRI spinal imaging reports were produced by board-certified radiologists and were written in English.

The SINS consists of six components (Table 1). The vertebral level with the highest total score was identified based on the individual SINS components. Pain history (mechanical, occasional, or absent) was extracted from patients’ clinical history in the electronic medical records (Epic Systems Corporation, Verona, WI, USA) and radiology request forms. The MRI reports were used to document the location of the lesion, vertebral body collapse, spinal alignment, and posterior element involvement (unilateral or bilateral). The CT report was used to evaluate bone lesion quality. To ensure an unbiased calculation of the SINS, impressions/conclusions from the MRI and CT reports, as well as any prior partial SINS calculations, were removed before the scoring process. The basis of calculating the SINS was restricted to only text from the radiological reports and no images were made available to the readers or large language models. This was to simulate common clinical practice whereby clinical notes remain the primary medium exchanged among clinicians in multidisciplinary tumour boards, across secure messaging platforms, or when there is a transfer of care between hospitals. An illustrated case example showing how the radiological reports were presented to the reference readers and residents is provided (Figure 1).

Three expert readers independently reviewed and calculated the SINS for all included reports, including two board-certified spine radiologists (J.H. with 14 years of experience and Y.L. with 4 years of experience) and one experienced spine surgeon (J.T. with 6 years of experience). Interobserver agreement was assessed among the expert readers. Subsequently, the expert readers convened to address any discrepancies to reach a final consensus SINS, which served as the reference standard for subsequent comparisons.

SINS calculations were also performed by two institutionally available privacy-preserving large language models (PP-LLMs)—Claude 3.5 (Anthropic PBC, San Francisco, CA, USA) and Llama 3.1 (Meta AI Research, Menlo Park, CA, USA). Additionally, two orthopaedic surgeons-in-training (G.T. and L.X.Y.; both in their second year of residency, which includes the completion of core spine oncology training and <18 months of supervised SINS reporting) independently calculated the SINS. These trainees were selected due to their active involvement in the clinical management and surgical referral process of patients with spinal metastases, and hence the frequent calculation of SINSs in their daily practice. 

Both the PP-LLMs and orthopaedic trainees were provided identical clinical information, including MRI and CT imaging reports, as well as clinical details from electronic medical records (EMRs) and imaging request forms. Their results were benchmarked against the consensus reference standard.

For the PP-LLMs, a standardised chain-of-thought prompt was developed to facilitate accurate SIN score calculation. The prompt guided the PP-LLMs through a structured, stepwise analysis, as follows:Extract the back pain history from the imaging request form and/or the latest clinical EMR entry.Determine lesion location, radiographic spinal alignment, vertebral body collapse, and posterior spinal element involvement from the MRI report.Assess the bone lesion quality based on the CT imaging report.

The PP-LLMs were conditioned to reproduce the six SINS categories with scoring rubrics per spinal level and to calculate the highest SINS per patient using this structured approach. A few-shot learning technique was utilised, whereby each PP-LLM was first trained on three complete worked examples with reference standard answers, demonstrating the intended reasoning process. No parameter fine-tuning or reinforcement learning was performed on either model. Both PP-LLMs were run in their frozen foundation states behind an institutional firewall. To ensure consistency and reliability, the models were explicitly instructed to avoid making up facts. The temperature parameter was set to 0 in order to minimise the randomness in the outputs.

### 2.2. Statistical Analysis

The primary outcomes of this study were as follows: First, to assess the agreement between clinician total SINS and the reference standard. Second, to assess the agreement between PP-LLM total SINS and the reference standard. Secondary outcomes were to assess the inter-rater agreement for individual SINS components and the overall accuracy of assigning SINS ratings (i.e., stable, indeterminate, or unstable based on total SINS).

The intraclass correlation coefficient (ICC) was used to measure the inter-rater agreement for the total SINS, while Gwet’s Kappa was used to measure the inter-rater agreement for the individual SINS components to account for the paradox effect caused by imbalanced distributions. Outcomes were interpreted according to the Landis and Koch scale, as follows: 0–0.20: slight agreement; 0.21–0.40: fair agreement; 0.41–0.61: moderate agreement; 0.61–0.80: substantial agreement; 0.81–1.00: almost perfect agreement [17]. Firstly, inter-rater agreement among the three expert readers was calculated to obtain a reference standard for the study. Next, inter-rater agreement was calculated between each clinician reader against the reference standard; then, the agreement between each PP-LLM and the reference standard was calculated. Inter-rater agreements were provided with corresponding 95% confidence intervals (CIs).

## 3. Results

A database of patients was available with known spinal metastatic disease who had attended a spinal surgery for a consultation. Overall, seven patients were excluded due to incomplete imaging data (no MRI and/or CT scan performed, or no written report available) and/or EMR entries documenting the history of pain. A total of 96 patients with 124 unique visits (each visit including a clinical entry, MRI, and CT spine) was available for analysis. The demographics of the included patients are summarised in Table 2. The mean patient age was 62 years (SD ± 10; range 32–86) and over half were female (53/96; 55.2%). The mean patient age for the females was 61 years (SD ± 11; range 32–86), and was 64 years (SD ± 9; range 40–83) for the males. Lung tumours were the most common primary tumour (37/96; 38.5%), followed by breast malignancies (16/96; 16.7%) and gastrointestinal tumours (10/96; 10.4%). The overall study design is summarised in Figure 2.

### 3.1. Reference Standard

Out of the 124 cases, 66 (53%) were classified as indeterminate, 38 (31%) as unstable, and 20 (16%) as stable. The inter-rater agreement for the total SINS and individual SINS components among all three expert readers is reported in Table 3. The agreement for the total SINS was almost perfect, with an ICC of 0.998 (*p* < 0.001; 95% CI [0.997, 0.999]). Agreement across individual SINS components was also almost perfect, with the lowest Gwet’s kappa for vertebral body collapse at 0.972 (*p* = 0.0138; 95% CI [0.945, 0.999]) and the highest for the location of the lesion at 1.000 (95% CI [1.000, 1.000]). The vertebral level identified with the highest SINS had an almost perfect agreement at 0.955 (*p* = 0.0154; 95% CI [0.925, 0.986]). Expert readers met to review scoring discrepancies and reached a consensus for each component of the SINS. Table 4 summarises the final distribution of the consensus SINS across the study population.

### 3.2. Overall Accuracy and Agreement for Total SINS

The accuracy of SINS ratings and inter-rater agreement for the total SINS between clinicians and PP-LLMs against the reference standard are reported in Table 5. The agreement among clinicians and the reference standard was almost perfect, with the lowest ICC of 0.926 for clinician 1 (*p* < 0.001; 95%CI [0.896, 0.948]) and the highest ICC of 0.986 for clinician 2 (*p* < 0.001; 95%CI [0.980, 0.990]). The agreement among LLMs and the reference standard was also almost perfect, with Claude 3.5 having a higher ICC at 0.984 (*p* < 0.001; 95%CI [0.978, 0.989]) compared to Llama 3.1, at 0.829 (*p* < 0.001; 95%CI [0.764, 0.877]). Compared to the clinician readers, Claude 3.5 had a higher ICC at 0.984 compared to clinician 1 at 0.926 (*p* < 0.001; 95%CI [0.896, 0.948]; *p* < 0.001). Llama 3.1 had an ICC of 0.829 and was outperformed by both clinician readers (*p* < 0.001). Overall, clinician 2 and Claude 3.5 had the highest accuracy of SINS assignment at 98.4% (122/124). Llama 3.1 had the lowest accuracy at 75% (93/124).

### 3.3. Subgroup Analysis for Individual SINS Components

The inter-rater agreement for individual SINS components between clinicians and PP-LLMs against the reference standard are reported in Table 6. There was at least a substantial agreement across all individual components between the clinicians and the reference standard (range: 0.702–0.979). Comparing clinician 1 against the reference standard, the lowest kappa was reported for pain at 0.702 (*p* < 0.001; 95%CI [0.604, 0.800]) and the highest kappa was reported for radiographic spinal alignment at 0.924 (*p* < 0.001; 95%CI [0.871, 0.976]). Similarly, the highest kappa for clinician 2 against the reference standard was reported for radiographic spinal alignment with an almost perfect agreement at 0.979 (*p* < 0.001; 95%CI [0.950, 1.000]). For the overall identification of the vertebral level with the highest SINS, both clinician 1 and clinician 2 had an almost perfect agreement, with kappas of 0.832 (*p* < 0.001; 95%CI [0.764–0.901] and 0.874 (*p* < 0.001; 95% CI [0.814, 0.935]), respectively.

Both PP-LLMs reported varying results when compared against the reference standard. For the Claude 3.5 model, there was an almost perfect agreement across all individual SINS components. The lowest kappa was reported for the location of the vertebral lesion at 0.919 (*p* < 0.001; 95%CI [0.864, 0.975]) and the highest kappa was reported for radiographic spinal alignment at 0.990 (*p* < 0.001; 95%CI [0.969, 1.000]). For the Llama 3.1 model, the lowest kappa was reported for bone lesion quality with fair agreement at 0.360 (*p* < 0.001; 95%CI [0.244, 0.476]). The highest kappa in the Llama 3.1 model was reported for radiographic spinal alignment, with substantial agreement at 0.744 (*p* < 0.001; 95%CI [0.653, 0.834]). For the overall identification of the vertebral level for the highest SINS, both the Claude 3.5 model and the Llama 3.1 model had an almost perfect agreement, with kappas of 0.950 (*p* < 0.001; 95%CI [0.910, 0.990]) and 0.874 (*p* < 0.001; 95% CI [0.813, 0.935]), respectively. There was no significant different for the overall identification of the vertebral level for the highest SINS between the clinicians and PP-LLMs (*p* > 0.05).

## 4. Discussion

This study evaluated the accuracy of two institutional privacy-preserving large language models (PP-LLMs)—Claude 3.5 and Llama 3.1—in calculating the Spinal Instability Neoplastic Score (SINS), using a consensus reference standard established by experienced spine oncology clinicians. Both PP-LLMs and clinician readers showed an almost perfect agreement with the reference standard for the total SINS. Between the two PP-LLMs, Claude 3.5 achieved the highest intraclass correlation coefficient (ICC 0.984; *p* < 0.001; 95% CI [0.978, 0.989]). This exceeded the performance of clinician 1 (ICC 0.926; *p <* 0.001; 95%CI [0.896, 0.948]) and Llama 3.1 (ICC 0.829; *p <* 0.001; 95% CI [0.764, 0.877]). The lowest ICC among the clinician readers was 0.926 (*p <* 0.001; 95% CI [0.896, 0.948]), indicating that the LLMs, particularly Claude 3.5, were able to approximate clinician-level scoring with a high degree of consistency. These findings demonstrate the potential of LLMs to assist with structured clinical tasks like the SINS, where consistency and efficiency are important. While this study appears to be the first to evaluate the accuracy of LLMs in the assessment of the SINS, the current literature has reported similarly promising results in other radiological scoring systems such as Coronary Artery Disease (CAD-RADS), Lung CT Screening Reporting and Data System (Lung-RADS), Ovarian-Adnexal Reporting and Data System (O-RADS), and Liver Imaging Reporting and Data (LI-RADS) [9,18,19,20]. Silbergleit et al. evaluated ChatGPT-3.5, ChatGPT-4o, Google Gemini, and Google Gemini Advanced against radiologist-assigned CAD-RAD scores. ChatGPT-4o demonstrated an almost perfect agreement (Cohen’s κ = 0.838), followed by Gemini Advanced, which has a substantial agreement (Cohen’s κ = 0.784) [9]. This is similar to the performance of the LLMs for the SINS, which demonstrated an overall almost perfect agreement in this study. Taken together, these results suggest a broader role for AI-driven tools in supporting clinical workflows that rely on standardised scoring and decision-making criteria.

Despite the almost perfect ICCs for total SINS among all clinicians and PP-LLMs, notable differences were observed in the agreement for individual components. Further subgroup analyses were performed to examine the accuracy of each SINS component. The Claude 3.5 model outperformed other readers, demonstrating an almost perfect agreement across all individual components, including the identification of the vertebral level with the highest total score (Gwet’s kappa = 0.950 (95% CI [0.910, 0.990)]. In contrast, the performance varied among the other readers, with the lowest agreement seen for pain [range: 0.455–0.702], vertebral body collapse [range: 0.518–0.763], and bone lesion quality [range: 0.360–0.884]. While clinician 2 showed an almost perfect agreement across all individual components [range: 0.919–0.990], clinician 1 had only a substantial agreement for the evaluation of pain (Gwet’s kappa = 0.702) and vertebral body collapse (Gwet’s kappa = 0.763). For the Llama 3.1 model, the lowest agreement was observed for bone lesion quality (Gwet’s kappa = 0.360) and pain (Gwet’s kappa = 0.404).

These findings are consistent with previous work by Fisher et al. [21] and Fourney et al. [22], who reported variable interobserver reliability across SINS components. Fourney et al. showed particularly low Fleiss κ values for bone lesion quality (κ = 0.244), spinal alignment (κ = 0.456), and vertebral body collapse (κ = 0.462) [22]. Similarly, Fisher et al. observed that interobserver agreement among radiation oncologists was substantial for the binary classification of stability (κ = 0.76) but was lower for individual components, particularly those reliant on subjective interpretation or limited documentation [15]. Fisher et al. also noted that the absence of image review and the reliance on narrative clinical records likely contributed to the variability in scoring. This issue is particularly relevant for the pain component, which can be difficult to categorise based on vague or inconsistently documented clinical descriptions that do not map clearly to the SINS classification (no pain, occasional pain, and mechanical pain) [21]. In addition, the accuracy of LLMs for the retrieval of clinical information has been reported to be approximately 67% in the current literature, which may contribute to errors in the computation of structured radiological scoring systems including the SINS [23].

There was also significant variation in performance between the two PP-LLMs. Claude 3.5 outperformed Llama 3.1 in both total SINS and across individual SINS components. Claude 3.5 achieved an almost perfect agreement with the reference standard across all components, including radiographic spinal alignment (Gwet’s κ = 0.990). Llama 3.1 had, at most, a substantial agreement for radiographic spinal alignment (Gwet’s κ = 0.744), and the lowest individual SINS component agreement was fair for bone lesion quality (Gwet’s κ = 0.360). Few studies have directly compared the performance of varied LLMs in healthcare applications, including the calculation of the SINS. However, it has been suggested that Claude 3.5 exhibits superior reasoning capabilities compared to other LLMs [24]. On the other hand, Llama 3.1 has been shown to be especially weak in complex problem-solving tasks [25]. Wu et al. evaluated the accuracy of ChatGPT-3.5, ChatGPT-4o, and Claude 2.0 in assigning scores for Lung-RADS, O-RADS, and LI-RADS using the same few-shot LLM training technique [18]. Although an earlier version of Claude was used, Claude 2.0 had the highest overall agreement (Fleiss κ = 0.66) and outperformed the other LLMs. The overall rating accuracy for Claude 2.0 in LI-RADS scoring was also the highest at 75% (45 correct ratings assigned out of 60 cases). This was comparable to the overall rating accuracy of the PP-LLMs evaluated in this study, with Claude 3.5 achieving the highest accuracy at 95% (122 correct ratings out of 124 cases) and Llama 3.1 at 75% (93 correct ratings out of 124 cases). In addition, when evaluating the performance of ChatGPT-3.5, Wu et al. reported moderate agreement (κ = 0.57) for overall RADS assignment (LI-RADS, Lung-RADS, and O-RADS) and a rating accuracy of 22.4% for LI-RADS assignment (22 out of 98 cases) [18]. These results are similar to another study by Fervers et al. (2024), which also evaluated the accuracy of ChatGPT-3.5 in assigning scores for LI-RADS [26]. That study reported a similar moderate agreement (κ = 0.44) and an overall rating accuracy of 44% (110/250) for ChatGPT-3.5 [26].

Similarly, two recent studies in gastroenterology demonstrated a better performance for Claude 3.5 over other models, including in diagnostic reasoning and the accuracy of clinical assessments [27,28]. Overall, the significant differences in LLM accuracy demonstrated in this study and outcomes from the existing literature suggest that the application of LLMs in healthcare should consider inherent architectural differences, as well as the strengths and weaknesses of each LLM. For example, the Claude 3.5 design focuses on advanced reasoning capabilities, making it adept at interpreting complex medical data and providing accurate diagnostic suggestions [29]. In contrast, the Llama 3.1 design is focused on an expansive architecture, which allows it to handle a broad range of tasks but may not be as finely tuned for medical reasoning tasks [30]. While acknowledging the transformative potential LLMs have in healthcare, studies from the broader machine learning literature such as the one from Cabitza et al. described concerns of unintended negative consequences when clinical workflows depend too heavily on machine learning systems, e.g., automation bias and uncertainty surrounding machine learning algorithms [31]. The rapid pace of development in LLMs adds further complexity, as newer and more capable models are continually emerging, making it challenging to establish consistent performance benchmarks or to confidently select models that are best suited for clinical use. Leevy et al. highlight that even current established methods such as oversampling and cost-sensitive learning can yield inconsistent outcomes in high-class-imbalanced healthcare data [32]. The SINS dataset distribution in this study classified 66/124 (53%) as indeterminate, 38/124 (31%) as unstable, and 20/124 (16%) as stable, demonstrating mild class imbalance. Since mild class imbalance could falsely inflate or deflate Cohen κ, Gwet’s AC1 was used to measure the inter-rater agreement for the individual SINS to avoid the paradox effect caused by imbalanced category frequencies. Additionally, more transparency needs to be established regarding the training process of the machine learning models used in healthcare. Roccetti et al. challenge the assumption that bigger training sets inherently lead to better outcomes, reinforcing the need for the careful curation and contextual evaluation of clinical datasets used in training machine learning models [33]. Overall, there is a need for rigorous and structured methods of comparing LLM performance in healthcare tasks to ensure the reliable and, most importantly, safe implementation of these tools. 

There were several limitations in this study. Firstly, only the main text body of the imaging reports was provided. The reference readers, clinician readers, and PP-LLMs did not have access to the actual imaging studies, which may have improved the accuracy of scores for some individual SINS components, as discussed above. However, the LLMs used would likely require training on imaging data (e.g., vision–language models) before they could be applied in this manner. This is an area of future research and would be synergistic with previously developed deep learning models for the closely related grading of epidural spinal cord compression [34,35,36]. In addition, a future multimodal vision–language project that fuses MRI and CT images with accompanying reports to provide an image-validated SINS is in process. Secondly, the evaluation of PP-LLM accuracy was conducted at a single time point; as such, little can be concluded about the reproducibility of the results. Thirdly, the study utilised only one prompt strategy (a few-shot prompting technique) to guide the PP-LLMs, and their performance may vary with alternative prompting methods. Fourthly, the dataset used in the training and evaluation of LLM performance in this study was confined to the cases available in the single study institution with mildly imbalanced category frequencies. Future work is underway to obtain external datasets from national and international collaborators to ensure broader generalizability. Fifthly, this study assessed LLM scoring performance alone, and future studies should be performed to evaluate potential time savings, workflow efficiency, and the impact on clinical decision-making. Finally, future work should also address ethical concerns such as model transparency and the development of regulatory frameworks to ensure safe deployment, protect patient data, and maintain compliance with clinical standards of care.

## 5. Conclusions

Our study demonstrates that LLMs, particularly Claude 3.5, exhibit a high accuracy for the calculation of the SIN score and may even outperform clinician readers. These findings highlight the potential role of advanced LLMs in augmenting clinical interpretation, particularly in structured scoring systems. Observed variations in LLM accuracy may be related to inherent architectural design and the prompt strategies used during the implementation of the LLMs. The performance variability between LLMs, in addition to notable discrepancies in scoring specific SINS components, underscores the need for further research to refine data inputs, optimise prompt engineering, and improve model alignment before routine clinical deployment.

## Figures and Tables

**Figure 1 cancers-17-02073-f001:**
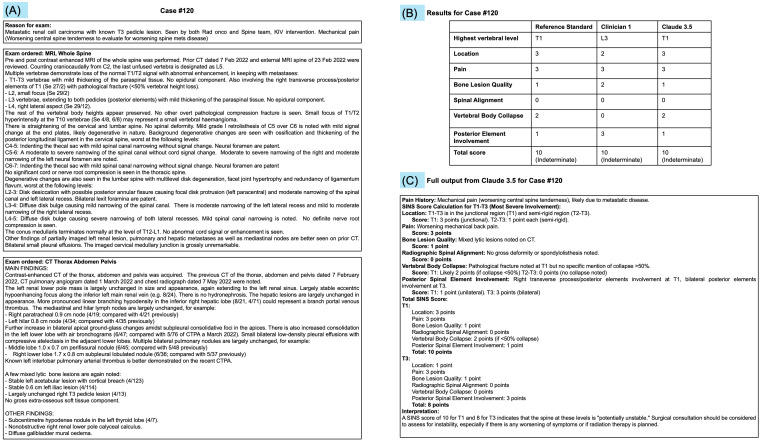
An illustrated case example demonstrating (**A**) how the reason for examination/clinical history, MRI, and CT radiological reports were shown to the readers; (**B**) tabulated results for this case example for the reference standard, clinician reader 1, and Claude 3.5; and (**C**) full output from the Claude 3.5 large language model.

**Figure 2 cancers-17-02073-f002:**
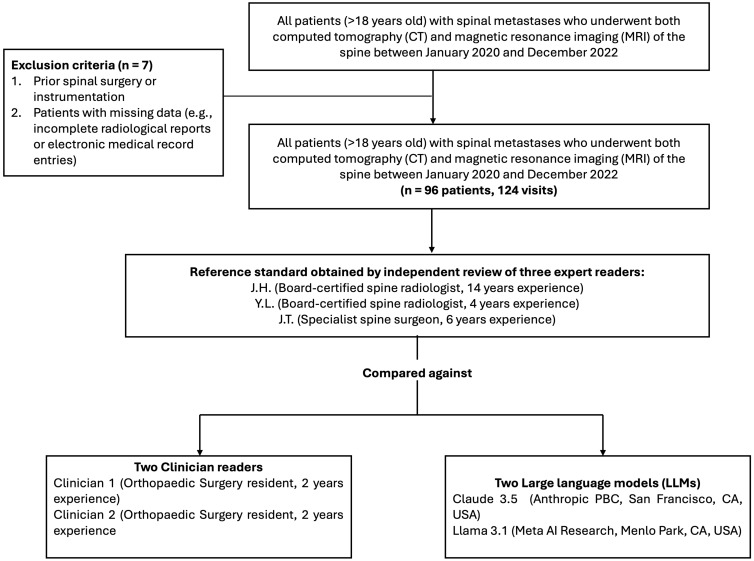
Summary of study methodology. Flow chart summarising the study design. Based on the inclusion and exclusion criteria, a total of 124 cases were included in this study. The 124 cases were reviewed by clinician readers and large language models and were compared against a reference standard set by expert readers.

**Table 1 cancers-17-02073-t001:** Spinal Instability Neoplastic Score.

Category	Score	Details
Location	3	Junctional (Occiput–C2; C7–T2; T11–L1; L5–S1)
	2	Mobile spine (C3–C6; L2–L4)
	1	Semirigid (T3–T10)
	0	Rigid (S2–S5)
Pain	3	Mechanical pain
	1	Occasional pain but not mechanical
	0	Pain-free
Bone Lesion Quality	2	Lytic
	1	Mixed lytic/blastic
	0	Blastic
Spinal Alignment	3	Subluxation/translation
	2	De novo deformity (kyphosis/scoliosis)
	0	Normal alignment
Vertebral Body Collapse	3	>50% collapse of vertebral body height
	2	<50% collapse of vertebral body height
	1	No collapse with >50% of vertebral body involvement
	0	None of the above
Posterior Element Involvement	3	Bilateral
	1	Unilateral
	0	None of the above
Interpretation of Total Score 0–6: Stable spine. 7–12: Indeterminate stability (further clinical evaluation may be needed). 13–18: Instability requiring surgical evaluation.		

**Table 2 cancers-17-02073-t002:** Demographic and oncological features of patients with spine metastases.

Demographics and Oncological Characteristics	Dataset (*n* = 96 Patients; MRI Spines: 124)
Mean age (years) for all patients	62 years (SD ± 10; range 32–86)
Sex	
Female	53/96 (55.2)
Male	43/96 (44.8)
Diagnosis	
Known Cancer	78/96 (81.2)
New Diagnosis	18/96 (18.8)
Cancer Type	
Lung	37/96 (38.5)
Breast	16/96 (16.7)
Gastrointestinal	10/96 (10.4)
Prostate	6/96 (6.3)
Gynaecologic	6/96 (6.3)
Renal	5/96 (5.2)
Myeloma/Plasmacytoma	5/96 (5.2)
Liver	4/96 (4.2)
Others	7/96 (7.3)

**Table 3 cancers-17-02073-t003:** Agreement of SINSs among expert readers.

	ICC ^1^	Gwet’s Kappa						
	Total Score	Location	Pain	Bone Lesion Quality	Radiographic Spinal Alignment	Vertebral Body Collapse	Posterior Spinal Element Involvement	Highest Vertebral Level
Inter-rater Agreement *	0.998 (0.997, 0.999)	1.000 (1.000, 1.000)	0.987 (0.968, 1.000)	0.976 (0.949, 1.000)	0.975 (0.949, 0.999)	0.972 (0.945, 0.999)	0.993 (0.979, 1.000)	0.955 (0.925, 0.986)
*p*—value	<0.001	0	0.0095	0.0139	0.0127	0.0138	0.007	0.0154

ICC: intraclass correlation coefficient. * Value in parentheses are 95% confidence intervals. ^1^ Single-measure ICC.

**Table 4 cancers-17-02073-t004:** SINS score distribution among patients included in the analysis.

SINS Range	Number of Cases (n)	Percentage (%)
0–6 (Stable)	20	16.1
7–12 (Indeterminate)	66	53.2
13–18 (Unstable)	38	30.6
Total	124	100%

**Table 5 cancers-17-02073-t005:** Overall agreement and accuracy between reference and clinician readers, as well as between reference and large language models for total SINS.

	ICC	*p*-Value	Overall Rating (Correct/Incorrect)	Percentage of Correct Ratings (%)
Reference—Clinician 1	0.926 (0.896, 0.948)	<0.001	110/14	88.7
Reference—Clinician 2	0.986 (0.980, 0.990)	<0.001	122/2	98.4
Reference—Claude 3.5	0.984 (0.978, 0.989)	<0.001	122/2	98.4
Reference—Llama 3.1	0.829 (0.764, 0.877)	<0.001	93/31	75

ICC: intraclass correlation coefficient.

**Table 6 cancers-17-02073-t006:** Agreement between reference and clinician readers, as well as between reference and large language models for individual SINS components.

Subcomponent	Reference—Clinician 1		Reference—Clinician 2		Reference—Claude 3.5		Reference—Llama 3.1	
	Gwet’s Kappa	*p*-Value	Gwet’s Kappa	*p*-Value	Gwet’s Kappa	*p*-Value	Gwet’s Kappa	*p*-Value
Location	0.899 (0.839, 0.960)	<0.001	0.910 (0.852, 0.968)	<0.001	0.919 (0.864, 0.975)	<0.001	0.455 (0.341, 0.568)	<0.001
Pain	0.702 (0.604, 0.800)	<0.001	0.965 (0.926, 1.000)	<0.001	0.954 (0.909, 0.999)	<0.001	0.404 (0.286, 0.522)	<0.001
Bone lesion quality	0.884 (0.814, 0.954)	<0.001	0.930 (0.875, 0.986)	<0.001	0.930 (0.875, 0.985)	<0.001	0.360 (0.244, 0.476)	<0.001
Radiographic spinal alignment	0.924 (0.871, 0.976)	<0.001	0.979 (0.950, 1.000)	<0.001	0.990 (0.969, 1.000)	<0.001	0.744 (0.653, 0.834)	<0.001
Vertebral body collapse	0.763 (0.672, 0.853)	<0.001	0.927 (0.874, 0.981)	<0.001	0.927 (0.873, 0.981)	<0.001	0.518 (0.399, 0.637)	<0.001
Posterior spinal element involvement	0.906 (0.845, 0.967)	<0.001	0.969 (0.933, 1.000)	<0.001	0.979 (0.950, 1.000)	<0.001	0.688 (0.587, 0.790)	<0.001
Highest vertebral level	0.832 (0.764, 0.901)	<0.001	0.874 (0.814, 0.935)	<0.001	0.950 (0.910, 0.990)	<0.001	0.874 (0.813, 0.935)	<0.001

ICC: intraclass correlation coefficient. Notes: values in parentheses are 95% confidence intervals. Clinicians 1 and 2 are both second-year residents in orthopaedic surgery.

## Data Availability

The data generated or analysed during the study are available from the corresponding author upon request.

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
