# Peer review of "Evaluating the Accuracy of Privacy-Preserving Large Language Models in Calculating the Spinal Instability Neoplastic Score (SINS)"

_cancers, 2025, doi:10.3390/cancers17132073_

Round 1

Reviewer 1 Report

Comments and Suggestions for Authors

The paper is rather well written, interesting and innovative. Nonetheless, there many obscure points that need clarification before considering it for publication.

It seems that this was the workflow:

1 Expert clinicians rated SINS scores from 124 MRI, CT reports and medical records, yielding a a kind of standard score

2 Two different LLMs were asked to score the same 124 reports

3 Two different human trainees were asked to score the same 124 reports

4 Scores from LLMS and human trainees were contrasted and compared with the standard scores

There is a very imprtant point here which is methodologically fundamental.

A Why should humans (experts and trainees) provide different scores. If this happens trainees have not been well trained.

B What was the training process that has led two general LLMs to learn how to provide SINS scores? If none explains how the LLMs have been trained to score SINS, what data, how many data,  ..., all this exercise remains a piece of magic without any relevance from a statistical viewpoint.

In conclusion, all these kind of papers cannot avoid to talk and reason and inform us about the data on which machines (and humans) have been trained. In absence of this we will never know if the training process has been valid enough to become general. For example waht about the dataset was an imbalanced one? In this sense, authors should read and cite the following:

- AA VV, Unintended consequences of machine learning in medicine, (2017) JAMA - Journal of the American Medical Association, 318 (6), pp. 517-518. doi: 10.1001/jama.2017.7797

- AA VV, Is bigger always better? A controversial journey to the center of machine learning design, with uses and misuses of big data for predicting water meter failures, J Big Data, Volume 6, Issue 11 2019, doi: 10.1186/s40537-019-0235-y

- AA VV, A survey on addressing high-class imbalance in big data, (2018) Journal of Big Data, 5 (1), doi: 10.1186/s40537-018-0151-6

Author Response

Reviewer #1’s Comment 1:

The paper is rather well written, interesting and innovative. Nonetheless, there many obscure points that need clarification before considering it for publication.

Author’s Response:

Thank you for taking the time for reviewing our manuscript. We appreciate your positive overall assessment. We have clarified each issue raised below.

Reviewer #1’s Comment 2:

It seems that this was the workflow: 1 Expert clinicians rated SINS scores from 124 MRI, CT reports and medical records, yielding a a kind of standard score. 2 Two different LLMs were asked to score the same 124 reports. 3 Two different human trainees were asked to score the same 124 reports. 4 Scores from LLMS and human trainees were contrasted and compared with the standard scores There is a very important point here which is methodologically fundamental.

  1. A) Why should humans (experts and trainees) provide different scores. If this happens trainees have not been well trained.

Author’s Response:

We agree that experience influences SINS scoring. We initially considered just comparing experts with the LLMs, but we also felt it was important to assess the performance of less experienced residents who are often tasked with calculating the SINS and determining the urgency of referral/treatment. The selected residents had completed only the core spine oncology training and accrued <18 months of supervised reporting. The difference in overall agreement among experts and residents mirrors the learning curve observed in daily practice. We have inserted the additional training details of the selected residents in the methodology section.

Reviewer #1’s Comment 3:

B What was the training process that has led two general LLMs to learn how to provide SINS scores? If none explains how the LLMs have been trained to score SINS, what data, how many data,  ..., all this exercise remains a piece of magic without any relevance from a statistical viewpoint.

Author’s Response:

We apologise for the ambiguity in our original description. No parameter fine‑tuning or reinforcement learning was performed on either model. Both Claude 3.5 (Anthropic) and Llama 3.1 (Meta) were run in their frozen foundation states behind an institutional firewall. We conditioned the models with a single system prompt that: (i) reproduced the six SINS categories with scoring rubrics per spinal level, (ii) instructed the model to extract the highest SINS per patient, and (iii) provided three complete worked examples with reference standard answers (few‑shot prompting). The temperature parameter was set to 0 in order to minimize the randomness in the outputs. No model weights were accessible or modified, ensuring compliance with our privacy‑preserving governance framework. We have further elaborated on the above details surrounding prompting and conditioning of the PP-LLMs in the methodology section.

Reviewer #1’s Comment 4:

In conclusion, all these kind of papers cannot avoid to talk and reason and inform us about the data on which machines (and humans) have been trained. In absence of this we will never know if the training process has been valid enough to become general. For example what about the dataset was an imbalanced one? In this sense, authors should read and cite the following:

- AA VV, Unintended consequences of machine learning in medicine, (2017) JAMA - Journal of the American Medical Association, 318 (6), pp. 517-518. doi: 10.1001/jama.2017.7797

- AA VV, Is bigger always better? A controversial journey to the center of machine learning design, with uses and misuses of big data for predicting water meter failures, J Big Data, Volume 6, Issue 11 2019, doi: 10.1186/s40537-019-0235-y

- AA VV, A survey on addressing high-class imbalance in big data, (2018) Journal of Big Data, 5 (1), doi: 10.1186/s40537-018-0151-6

Author’s Response:

We agree that dataset distribution and external validity deserve further discussion. Of the 124 cases, 66 (53 %) were classified as indeterminate, 38 (31 %) as unstable, and 20 (16 %) as stable according to the reference standard. This mild class imbalance can inflate or deflate Cohen κ; therefore, Gwet’s Kappa (AC1) was used to measure inter-rater agreement for the individual SINS components to avoid the paradox effect caused by imbalanced category frequencies. We were constrained by the cases available at our institution, and future work is already under way to obtain external datasets from national and international collaborators to ensure broader generalizability. This constraint has been highlighted in the limitations section. The revised manuscript now cites the three seminal papers recommended by the reviewer to frame this issue in an expanded discussion: Cabitza et al. (2017), Roccetti et al. (2019), and Leevy et al. (2018).

We trust that these revisions address all concerns and substantially improve the manuscript.

Reviewer 2 Report

Comments and Suggestions for Authors

Authors present retrospective analysis of 124 radiology reports of patients with spinal metastases, the accuracy of two institutional privacy-preserving 
LLMs (large language models), Claude 3.5 and Llama 3.1, in computing instability scoree SINS from radiology reports and electronic
medical records, comparing their performance against clinician readers. Both LLMs and clinicians  
demonstrated almost perfect agreement with the reference standard for total SINS; Claude outperformed LLM. One major issue with this study, which also needs to be acknowledged in the limitatation, is that these results are based solely on radiological reports - clinical spine surgeons usually do not need radiological reports for assessment of SINS. Analysis of imaging would have make more sense for SINS evaluation. Also, for better overview of materials and methods, I suggest to include an illustrative case - how did the report look and how did the resident, according to report, calculated SINS. 

Author Response

Reviewer #2’s Comment 1:

Authors present retrospective analysis of 124 radiology reports of patients with spinal metastases, the accuracy of two institutional privacy-preserving LLMs (large language models), Claude 3.5 and Llama 3.1, in computing instability score SINS from radiology reports and electronic medical records, comparing their performance against clinician readers. Both LLMs and clinicians demonstrated almost perfect agreement with the reference standard for total SINS; Claude outperformed LLM. One major issue with this study, which also needs to be acknowledged in the limitation, is that these results are based solely on radiological reports - clinical spine surgeons usually do not need radiological reports for assessment of SINS. Analysis of imaging would have make more sense for SINS evaluation.

Author’s Response:

Thank you for taking the time to review our paper, and for highlighting this limitation. We intentionally restricted the study to text‑based SINS extraction because radiology reports and clinical notes remain the primary medium exchanged in tumour boards and secure healthcare messaging platforms and are often the only documentation carried forward when patients are transferred between hospitals. We have added this justification to the methodology and acknowledged the limitation in the discussion. In addition, we now describe an ongoing multimodal vision–language project that fuses MR and CT images with accompanying reports to provide image‑validated SINS in future deployments.

Reviewer #2’s Comment 2:

Also, for better overview of materials and methods, I suggest to include an illustrative case - how did the report look and how did the resident, according to report, calculated SINS.

Author’s Response:

Thank you for the suggestion and we agree this will be a great addition to the manuscript. We have added Figure 1 that contains (a) a de‑identified excerpt of a combined MRI & CT report, and (b) the level‑by‑level SINS calculated by Resident #1, Claude 3.5 and the reference standard. This example is referenced in Methods and improves the transparency of our workflow.

We trust that these revisions address all concerns and substantially improve the manuscript.

Round 2

Reviewer 1 Report

Comments and Suggestions for Authors

I think that, even if not all my answers have been completely responded, the paper has increased in quality after revision and deserves publication now.